# New Propargyloxy Derivatives of Galangin, Kaempferol and Fisetin—Synthesis, Spectroscopic Analysis and In Vitro Anticancer Activity on Head and Neck Cancer Cells

**DOI:** 10.3390/cells12182288

**Published:** 2023-09-15

**Authors:** Robert Kubina, Kamil Krzykawski, Arkadiusz Sokal, Marcel Madej, Arkadiusz Dziedzic, Monika Kadela-Tomanek

**Affiliations:** 1Department of Pathology, Faculty of Pharmaceutical Sciences in Sosnowiec, Medical University of Silesia in Katowice, Ostrogórska 30, 41-200 Sosnowiec, Poland; 2Silesia LabMed, Centre for Research and Implementation, Medical University of Silesia in Katowice, 18 Medyków Str., 40-752 Katowice, Poland; kamil.krzykawski@sum.edu.pl (K.K.); marcel.madej@sum.edu.pl (M.M.); 3Students Scientific Group of Department of Organic Chemistry, Faculty of Pharmaceutical Sciences in Sosnowiec, Medical University of Silesia, 41-200 Sosnowiec, Poland; s77856@365.sum.edu.pl; 4Department of Conservative Dentistry with Endodontics, Medical University of Silesia, 40-055 Katowice, Poland; adziedzic@sum.edu.pl; 5Department of Organic Chemistry, Faculty of Pharmaceutical Sciences in Sosnowiec, Medical University of Silesia, 41-200 Sosnowiec, Poland; mkadela@sum.edu.pl

**Keywords:** flavonol, anticancer, flow cytometry, apoptosis, head and neck cancer, cyclin

## Abstract

Head and neck cancer (HNC) therapy is limited; therefore, new solutions are increasingly being sought among flavonoids, which exhibit numerous biological properties, including potential anticancer activity. However, because they are mostly insoluble in water, are unstable and have low bioavailability, they are subjected to chemical modification to obtain new derivatives with better properties. This study aimed to synthesize and analyze new propargyloxy derivatives of galangin, kaempferol and fisetin, and to evaluate their anticancer activity against selected HNC cell lines. The obtained derivatives were assessed by spectroscopic analysis; next, their anticancer activity was evaluated using a flow cytometer and real-time cell analysis. The results showed that only the fisetin derivative was suitable for further analysis, due to the lack of crystal formation of the compound. The fisetin derivative statistically significantly increases the number of cells in the G2/M phase (*p* < 0.05) and increases cyclin B1 levels. A statistically significant increase in the number of apoptotic cells after being exposed to the tested compound was also observed (*p* < 0.05). The data indicate that the obtained fisetin derivative exhibits anticancer activity by affecting the cell cycle and increasing apoptosis in selected HNC lines, which suggests its potential use as a new medicinal agent in the future.

## 1. Introduction

The seventh most common type of cancer globally is head and neck cancer (HNC) which refers to a diverse group of cancers originating from the anatomical areas that make up the upper gastrointestinal track [1]. Squamous cell carcinoma of the oral cavity, pharynx and larynx (SCC) accounts for almost 90% of cases. Surgical resection, radiotherapy or a combination of the two have been used as treatment options over the past few decades [2]. Head and neck squamous cell carcinoma (HNSCC) exists in the form of a very challenging tumor. Despite developments in treatment, the overall prognosis for HNSCC remains poor, with a five-year survival rate of approximately 50%. Due to the high complexity and heterogeneity of HNSCC, the response to treatment methods is highly variable, in addition to having a range of clinicopathological risk factors. The heterogeneity of HNSCC consists of several factors, including mutations in the genetic materials, the presence of the tumor microenvironment and the associated immune response [3]. Anatomically, the four main areas involved in HNC are the oral cavity, pharynx, nasal cavity and larynx. Histologically, HNC starts with oral precancerous lesions such as leukoplakia, erythroplakia and oral submucosal fibrosis and malignancies such as squamous cell carcinoma, adenocarcinoma, lymphoma and blastoma [4].

In order to combat HNC, dynamic compounds directed at multiple targets that can block and/or aim at different signaling pathways need to be identified. Platinum-based chemotherapy with cetuximab is the gold standard treatment for patients with recurrent disease [5]. The currently approved first-line therapy for HNC patients including surgery, chemotherapy (docetaxel, paclitaxel, carboplatin, cisplatin, 5-fluorouracil and gemcitabine), as well as molecularly targeted agents such as nivolumab, pembrolizumab and cetuximab [6]. Phytochemicals fit this scenario as they are a rich source of such compounds aimed at multiple targets. One reason for the efficacy of phytochemicals in the prevention and treatment of HNC is their direct contact with oral tissues before they are absorbed and metabolized in the gastrointestinal tract [4].

Natural products are an important starting point for drug development. Numerous natural products and their derivatives have been approved by the Food and Drug Administration (FDA) or are in clinical trials, such as artemether, lovastatin, curcumin and resveratrol [7].

Evidence on the pharmacological properties of phytochemicals suggests that, in addition to their anti-inflammatory, immunomodulatory and antioxidant effects, they may modulate various processes and signaling pathways involved in carcinogenesis. Flavonols are thought to effect risk reduction via chemopreventive mechanisms of action that regulate the cell cycle, proliferation and cell death by apoptosis and contribute to the modulation of the metabolism of carcinogen and related inflammatory pathways [8,9].

In addition, flavonols exhibit anticancer effects through the inhibition of tumor cell angiogenesis and proapoptotic effects. These anticancer properties have been demonstrated in several cancer cell lines such as: RAW264.7; U2OS; MCF7; MDA-MB-231; Caco-2; HT-29; HCT116; HepG2; A549; Lu-1; SKOV-3; LNCaP; Du-145; HeLa; HSC-3; SCC-9; and SCC-25, in addition to antioxidant effects [10]. The antioxidant effects of flavonol glycosides are weaker than those of flavonol aglycones [11].

Due to the lack of effective and safe therapies, cancer still remains a fatal disease in most cases. However, chemotherapy is in fact still the most common form of treatment, but in a large group of patients, the cells can develop resistance. Some of the crucial problems of currently applicable anticancer drugs are their systemic toxicity and relatively low selectivity towards cancer cells, which often leads to multidrug resistance. Recently, reports of the discovery of new medicinal substances of natural origin, with fewer side effects and that are more cost-effective, have become increasingly frequent.

The modification of natural products and their derivatives was expected to be essential for the design of new drugs. Studies have suggested that the backbone structure of flavonoids is an important factor in their activity and that hydroxylation is a critical process in assessing their activity [12,13].

Alkyne compounds that contain at least one triple carbon–carbon bond are widely distributed in nature. Marine organisms such as cyanobacteria, algae and sponges are rich sources of these compounds. Moreover, the alkyne substance has been isolated from herbs and bacteria. The alkyne substance exhibits high biological activities such as anticancer, antibacterial and antiviral [14,15,16,17]. The introduction of an alkyne moiety into compounds influences the biological and physicochemical properties of the obtained compound [18,19,20,21,22].

The aim of this work was to synthesize and analyze new propargyloxy derivatives of galangin, kaempferol and fisetin and evaluate their anticancer activity against selected HNC cell lines.

## 2. Materials and Methods

### 2.1. Materials

All commercial reagents, solvents and other chemicals were used without further purification. Fisetin (ref. No. PHL82542), galangin (ref. No. 92342), kaempferol (ref. No. 60010) and other chemicals and solvents were purchased from Merck (Darmstadt, German). Dulbecco’s modified Eagle’s medium (DMEM), McCoy’s medium, Eagle’s minimum essential medium (EMEM), Ham’s nutrient mixture F-12, Trypsin EDTA solution and fetal bovine serum (FBS) were purchased from Gibco (Gran Island, NY, USA). Cal-27, FaDu, Detroit 562, UCST-OT-1109 and A-253 cell lines were obtained from the American Type Culture Collection (ATCC, Manassas, VA, USA). The Beas-2B cell line was obtained from Sigma Aldrich (Darmstadt, Germany).

### 2.2. Chemistry

The nuclear magnetic resonance (NMR) spectra were measured using the Bruker Avance 600 spectrometer (Bruker, Billerica, MA, USA) in acetone-d6 solvents. Chemical shifts (δ) are reported in ppm and J values in Hz. Multiplicity is designated as singlet (s), doublet (d), triplet (t) and multiplet (m). High-resolution mass spectral analysis (HR-MS) was recorded using the Bruker Impact II instrument (Bruker, Billerica, MA, USA). The theoretical molecular weight calculations of compounds were performed using “The Exact Mass Calculator, Single Isotope Version” tool (http://www.sisweb.com/referenc/tools/exactmass.htm, accessed on 1 June 2023) (Ringoes, NJ, USA). Infrared (IR) spectra (Shimadzu, Kyoto, Japan) were performed using the IRXross spectrophotometer equipped with the attenuated total reflection (ATR) mode. Melting points were measured by the Electro-thermal IA 9300 (Electrothermal, London, UK) melting point apparatus.

#### General Synthesis of Compounds **4**–**6** 

Compounds **1**–**3** (1 mmol) and potassium carbonate (5 eq, 5 mmol, 0.69 g) were suspended in acetone (5 mL). The propargyl bromide (5 eq, 5 mmol, 0.595 g) was added drop by drop and the mixture was stirred at reflux. The reaction progress was monitored using the TLC method. After 24 h, the reaction mixture was concentrated using a vacuum evaporator. The crude product was purified by column chromatography (SiO2, chloroform/ethanol, 15:1, *v*/*v*) to obtain pure compounds **4**–**6**.

5-hydroxy-2-phenyl-3,7-bis(prop-2-yn-1-yloxy)-4H-chromen-4-one 4: Yield 60%; ^1^H NMR (600 MHz, acetone-d6) δ, ppm: 2.89 (t, J = 2.4 Hz, 1H), 3.09 (t, J = 2.4 Hz, 1H), 4.84 (d, J = 2.4 Hz, 2H), 4.89 (d, J = 2.4 Hz, 2H), 6.31 (d, J = 2.4 Hz, 1H), 6.66 (d, J = 2.4 Hz, 1H), 7.45–7.49 (m, 3H), 8.03–8.04 (m, 2H) and 12.41 (s, 1H) (Appendix A); ^13^C NMR (150 MHz, acetone-d6) δ, ppm: 56.2, 59.1, 77.1, 77.4, 77.8, 78.3, 93.2, 98.5, 106.0, 128.5, 128.8, 130.5, 131.1, 136.8, 156.9, 157.0, 161.9, 163.8 and 178.8 (Appendix A); IR (ν^max^ cm^−1^, ATR): 3275, 3246, 2118, 1649, 1605, 1449 and 1167; ESI-HRMS *m*/*z* [M+Na]+ calcd for C_21_H_14_O_5_Na^+^ 369.0739, found 369.0738.

5-hydroxy-3,7-bis(prop-2-yn-1-yloxy)-2-(4-(prop-2-yn-1-yloxy)phenyl)-4H-chromen-4-one 5: Yield 64%; ^1^H NMR (600 MHz, aceton-d6) δ, ppm: 2.87 (t, J = 2.4 Hz, 1H), 3.03 (t, J = 2.4 Hz, 1H), 3.07 (t, J = 2.4 Hz, 1H), 4.79 (d, J = 2.4 Hz, 2H), 4.81 (d, J = 2.4 Hz, 2H), 4.87 (d, J = 2.4 Hz, 2H), 6.28 (d, J = 2.4 Hz, 1H), 6.64 (d, J = 2.4 Hz, 1H), 7.04–7.05 (m, 2H), 8.06–8.07 (m, 2H) and 12.45 (s, 1H) (Appendix A); ^13^C NMR (150 MHz, acetone-d6) δ, ppm: 55.6, 56.2, 59.0, 76.7, 77.1, 77.3, 77.9, 78.3, 78.4, 93.3, 98.4, 105.9, 114.8, 123.4, 130.5, 131.1, 136.1, 156.7, 156.8, 159.9, 161.9, 163.6 and 178.6 (Appendix A); IR (ν_max_ cm^−1^, ATR): 3285, 3264, 2118, 1655, 1597, 1493 and 1175; ESI-HRMS *m*/*z* [M+Na]+ calcd for C_24_H_16_O_6_Na^+^ 423.0844, found 423.0845.

2-(3,4-bis(prop-2-yn-1-yloxy)phenyl)-3,7-bis(prop-2-yn-1-yloxy)-4H-chromen-4-one 6: Yield 58%; ^1^H NMR (600 MHz, acetone-d6) δ, ppm: 2.87 (t, J = 2.4 Hz, 1H), 3.03 (t, J = 2.4 Hz, 2H), 3.08 (t, J = 2.4 Hz, 1H), 4.78 (d, J = 2.4 Hz, 2H), 4.82 (d, J = 2.4 Hz, 2H), 4.87 (d, J = 2.4 Hz, 2H), 4.97 (d, J = 2.4 Hz, 2H), 7.00–7.01 (m, 1H), 7.16–7.17 (m, 2H), 7.73–7.75 (m, 1H), 7.91–7.95 (m, 1H) and 7.96 (d, J = 9.0 Hz, 1H) (Appendix A); ^13^C NMR (150 MHz, acetone-d6) δ, ppm: 56.2, 56.7, 58.5, 76.7, 76.8, 77.1, 77.8, 78.4, 78.6, 78.8, 101.7, 113.8, 114.7, 115.6, 118.2, 122.9, 124.3, 126.6, 137.8, 147.1, 149.8, 155.2, 156.6, 162.1 and 173.3 (Appendix A); IR (ν_max_ cm^−1^, ATR): 3294, 3242, 2924, 2855, 2118, 1620, 1597, 1437 and 1176; ESI-HRMS *m*/*z* [M+Na]+ calcd for C_27_H_18_O_6_Na^+^ 461.1001, found 461.1006.

### 2.3. Derivatives Treatment

The tested derivatives stock solutions (5 mg/mL) were prepared for further analysis by dissolving in dimethyl sulfoxide. Stocks were stored at −80 °C until their use (no longer than two weeks). To perform the analysis, the concentrations of the test compounds ranged from 1 to 40 μM. The control cells were cultured in the appropriate culture medium with DMSO without the addition of the tested derivatives.

### 2.4. Cell Culture

A253 (submaxillary salivary gland carcinoma, male, ATCC HTB-41) cells were cultured in McCoy’s medium; FaDu (pharynx squamous cell carcinoma, male, ATCC HTB-43) and Detroit 562 (pharynx carcinoma derived from pleural fluid, female, ATCC CCL-138) were grown in Eagle’s minimum essential medium (EMEM); UCSF-OT-1109 (tongue squamous cell carcinoma, female, CRL-3442) and CAL 27 (tongue squamous cell carcinoma, male, CRL-2095) were cultured in Dulbecco’s modified Eagle’s medium (DMEM). All media were supplemented with 10% fetal bovine serum (FBS), 100 μg/mL penicillin and 100 μg/mL streptomycin. Beas-2B (normal bronchial epithelium, cod. 95102433) cells were cultured in Ham’s nutrient mixture F-12 with 10% fetal bovine serum. All cells were maintained at 37 °C in a humidified atmosphere (95%) containing 5% CO_2_. Cells were seeded overnight and then incubated with various concentrations of a kaempferol derivative, galangin derivative and fisetin derivative for 24 and 48 h. For each concentration and time course study, there was a control sample that remained untreated (medium+DMSO without derivatives) and received an equal volume of medium. All different treatments were carried out in triplicate.

### 2.5. Cell Cytotoxicity (IC50 Determination)

Cell viability was determined using a modified MTT assay (Merck, Darmstadt, German). Shortly, cells were seeded (5000 cells/well) into 96-well culture plates and allowed to grow for 24 h and then treated with kaempferol, galangin and fisetin derivatives (0, 1.3, 5, 10, 15, 20, 25, 30, 35 and 40 μM) for 24 and 48 h. The MTT assay was used to measure cytotoxicity. After the removal of the medium, cells were labelled with MTT solution (1 mg/mL in medium) for 4 h and the resulting formazan dye was dissolved in DMSO (200 μL). Absorbance was measured at 570 nm (650 nm as reference) in a Varioskan Lux ELISA plate reader (Thermo Scientific, Waltham, MA, USA).

### 2.6. Real-Time and Dynamic Monitoring of Cell Proliferation Using the xCELLigence RTCA

For all impedance experiments, the xCELLigence RTCA instrument (Agilent, Santa Clara, CA, USA) was used together with RTCA-DP software. The growth of cells on the electrode surface over time leads to changes in the recorded impedance represented as a Cell Index value, the increase is proportional to the number of adherent cells. Cells were seeded into each well at a density of 7.0 × 10^3^ cells/well. Cell growth recording was started immediately after plate preparation. Fisetin derivatives at a concentration of ½ IC_50_ and ¼ IC_50_ were added after 24 h. After that, impedance changes were monitored every 15 min for the next 48 h. Incubation was carried out at 37° with 5% CO_2_ in a volume of 200 µL.

### 2.7. Cell Cycle Detection Assay

After the cells reached 60–70% confluence as assessed by JuLi Br (NanoEnTek, Seoul, Korea), the growth medium was replaced with a medium containing different doses of fisetin derivative (equivalent to IC_50,_ ½ IC_50_ and ¼ IC_50_ for each cell line, respectively). After 24 h of incubation, cells were washed with PBS and fixed by adding 70% ethanol and incubated for at least 2 h at −20 °C. Next, after removing ethanol, the cells were stained by adding 0.5 mL of the PI solution containing RNase and incubated at room temperature (RT) in the dark for 15 min. DNA content was then analyzed using a CytoFlex Srt flow cytometer (Beckman Coulter, Miami, FL, USA). The data were analyzed using the CytExpert Srt software version 1.1. (Beckman Coulter, Miami, FL, USA).

### 2.8. Cyclins D1 and B1 Analysis

Cells were seeded into cell culture flasks (T75) (5 × 10^5^ cells). On the second day, the medium was changed, and the cells were treated with a fisetin derivative. Cells were incubated for 24 h before harvesting. The cells were washed with a wash buffer and gently fixed by adding cold 75% ethanol (cyclin B1) or 75% methanol (cyclin D1) before being placed in a freezer for min. 2 h. Next, ethanol/methanol was removed, and the cells were treated with 0.25% Triton X-100 for 5 min in an ice bath. The cells were washed and resuspended in a wash buffer at a final concentration of 1 × 10^7^ cells/mL. Then, 100 μL of obtained cell suspension was transferred to new tubes and stained by adding 20 μL of FITC cyclin B1 or FITC cyclin D1 and incubated in the dark for 30 min at RT. After 30 min of incubation, the cells were washed and resuspended in 0.5 mL 7-AAD solution and incubated for 20 min at 4 °C. Then, the cells were analyzed using a CytoFlex Srt flow cytometer. For each measurement, at least 10,000 cells were counted.

### 2.9. Cell Apoptosis Assay

Cell apoptosis was evaluated via flow cytometry by staining cells with FITC Annexin V in combination with propidium iodide (PI) (BD Bioscience, San Diego, CA, USA). After a 24 h treatment with a fisetin derivative, cells floating in the medium were collected. The adherent cells were detached using 0.25% trypsin in EDTA. The culture medium containing 10% FBS was added to inactivate trypsin. Then, the cells were washed twice with cold PBS and resuspended in 1X Binding Buffer at a concentration of 1 × 10^6^ cells/mL. Next, 100 μL of obtained cell suspension was transferred to new tubes and stained by adding 5 μL PI and FITC Annexin V. After 15 min of incubation at RT in the dark, 400 μL of 1X Binding Buffer was added to each tube and the cells were analyzed immediately after staining using a CytoFlex Srt flow cytometer. Untreated cells were used as the control for double staining.

### 2.10. Statistical Analysis

Data are presented as means ± standard deviation (SD). Statistically significant differences were analyzed by Student’s *t*-test. Statistical analysis was performed using GraphPad Prism V software (prism 7.05 (windows)). *p*-value < 0.05 was considered statistically significant.

## 3. Results

### 3.1. Chemistry

The propargyloxy derivatives 4–6 were obtained in the reaction of flavonol 1–3 with the propargyl bromide in the presence of potassium carbonate (K_2_CO_3_) and acetone as a solvent (Figure 1).

After the reaction, the products were purified via column chromatography. Pure products 4–6 were obtained with a yield in the range of 58–64%. The chemical structure of compounds **4**–**6** was confirmed by ^1^H and ^13^C NMR, IR and HR-MS spectra. In addition, to confirm the correct substitution in derivatives 4–5, the Heteronuclear Single Quantum Coherence (HSQC) and Heteronuclear Multiple Bond Correlation (HMBC) spectra were measured (Appendix A). Table 1 and Table 2 show a proton–carbon correlation for compounds **4** and **5**, respectively.

In the ^13^C NMR spectrum of compound **4**, the signal at 178.8 ppm was assigned to the carbon atom at the C4 position. Based on the HMBC correlation spectra, the signal at 6.66 ppm and 6.32 ppm was assigned to H8 and H6 protons, respectively. The HSQC spectra show that the chemical shift of the carbon atom at the C6 position is equal to 98.4 ppm. The HMBC spectrum was used to identify the location of the propargyloxy substituent in the flavone moiety. The hydroxyl group was assigned to the C5 position in the flavone moiety based on the correlation between the proton signal at 12.41 ppm with carbon atoms at C6 (δC 98.4 ppm), C7 (δC 106.0 ppm) and C10 (δC 161.9 ppm) positions. The methylene group (δH 4.89 ppm) at the C11 position correlated with carbon atoms at the C2 (δC 136.8 ppm) and C12 (δC 78.3 ppm) positions. Furthermore, the methine group (δH 2.88 ppm) at the C13 position correlates with the carbon (δC 59.1 ppm) atom at the C11 position. The chemical shift of proton and carbon atoms of propargyloxy substituent at the C3 position was also assigned based on the HMBC spectrum (Table 1).

A similar analysis was carried out for kaempferol derivatives 5 (Table 2).

The position of the hydroxyl group was established basis of the correlation between the proton at 12.45 ppm with carbon atoms at C4 (δC 178.6 ppm), C6 (δC 98.4 ppm), C7 (δC 105.9 ppm) and C10 (δC 161.9 ppm) positions (Table 2). The HSQC and HMBC spectra analysis was also used to determine the chemical shift of the hydrogen and carbon atoms at propargyloxy groups (Appendix A). The proton (δH 4.87 ppm) at C11 position correlated with carbon atoms at C2 (δC 136.1 ppm) and C12 (δC 78.3 ppm) position. The correlation between C11 (δC 59.0 ppm) and H13 (δH 2.87 ppm) was also observed. The correlation between the signal of the carbon atom at C4′ (δC 159.9 ppm) and the signal at 4.79 ppm allows the arrangement of this signal to the methylene group at the C17 position.

### 3.2. Fisetin Derivative Inhibits Cell Viability and Proliferation of HNCs

All propargyloxy derivatives obtained were evaluated for their in vitro cytotoxicity against five human head and neck region cancer cell lines and one normal bronchial epithelial cell (Beas-2B) using the MTT assay (Figure 2A). It was found that only the fisetin derivative could be analyzed in detail for its mechanism of action. Kemferol and galangin derivatives led to precipitation of crystals thereby making the results obtained unreliable. Therefore, a fisetin derivative was used for all further steps. This compound shows high cytotoxicity against four cancer cell lines, with IC_50_ values of 5.20, 8.50, 8.64 and 8.17 μM for the Cal-27, FaDu, Detroit 562 and UCST-OT-1109 lines, respectively. For the A-253 line, the IC_50_ value was 28.24 μM, which was more than three times higher than for the other tested lines. The next step was to determine the selectivity index (SI). For this purpose, normal cells of the Beas-2B line were analyzed. The selectivity index can be defined as the ratio of the concentration of a toxic compound to its effective bioactive concentration. To evaluate the anticancer activity of a compound, its cytotoxicity against non-malignant cell lines must be determined to calculate the SI value with the formula:*SI* = *IC*_50_
*no cancer cells*/*IC*_50_
*cancer cells*

Weerapreeyakul et al. [23] proposed an *SI* value (≥3) to evaluate the anticancer activity of an active substance. A low *SI* (<1) means that the compound may be toxic and cannot be used as a drug.

In addition, the antiproliferative activity of the fisetin derivative was evaluated against tumor cells in HNCs using the xCELLigence^®^ RTCA instrument, which uses non-invasive electrical impedance monitoring to quantify cell proliferation. As shown in Figure 2B, the fisetin derivative inhibited the proliferation of all cell lines tested. The greatest antiproliferative effect was observed for the Cal-27 line at both ½ IC_50_ and ¼ IC_50_ concentrations, which correlates with the results obtained in the MTT assay, where the IC_50_ value for the Cal-27 line was the lowest.

### 3.3. Fisetin Derivative Induces Cell Cycle Arrest in HNC Cell Lines

Since treatment with a fisetin-derivative-inhibited the proliferation of cancer cells, in the tongue, submaxillary salivary gland and pharynx, we decided to determine whether this was due to the induction of cell cycle arrest. The role of the studied flavonols in cell cycle arrest was analyzed via flow cytometry using a cell cycle phase detection kit (Figure 3). The sub-G1 phase is a measure of late cell apoptosis, the G0/G1 phase is the resting state of cells, and the S and G2/M phases express the percentage of cells in the proliferation phase. An increase in the percentage of cells in any phase of the cell cycle under the influence of the applied drug means that the progression of the cycle in this phase was inhibited. The consequence of disruption of the normal course of the cell cycle is cell apoptosis. The data obtained show that the fisetin derivative decreased the population of cells in the G0/G1 phase and correspondingly increased the population of cells in the G2/M phase, which was particularly evident in the case of cells of the A-253, Cal-27 and UCSF-OT-1109 lines. At the highest concentration of IC_50_, an increase in cells in the G2/M phase was observed from 22% to 32.70%, 12.20% to 23.97% and 15.32% to 29.41% for cells of the A-253, Cal-27 and UCSF-OT-1109 lines, respectively. For the ½ IC_50_ concentration, cycle inhibition in the G2/M phase was observed for cells of the A-253 and UCSF-OT-1109 lines and was 34.37% and 32.31%, respectively. The lowest test concentration of the fisetin derivative also had an effect on the cell cycle by inhibiting it in the G2/M phase for the A-253 and UCSF-OT-1109 lines. Interestingly, statistically significant changes were also observed in the S phase of the cell cycle for cells of the Cal-27 line at both IC_50_ and ½ IC_50_ concentrations. The increase was 23.15% and 22.64%, respectively, relative to the control, i.e., 14.14%. In addition, the appearance of cells in the sub-G1 phase of the cell cycle was observed for the A-253 line at IC_50_ and FaDu at IC_50_ and ½ IC_50_. These changes were statistically significant. No statistically significant changes in cell cycle distribution were observed for the Detroit 562 line.

### 3.4. Fisetin Derivative Treatment Increases the Steady State of Cyclin B1 Protein

As we have shown, the propargyloxy derivative of fisetin decreased the cell population in the G0/G1 phase and correspondingly increased the cell population in the G2/M phase, which was particularly evident in cells of the A-253, Cal-27 and UCSF-OT-1109 lines, we decided to see if it affects the amount of cyclin B1 and cyclin D1 in these cell lines. Cyclin B1 is one of the important factors in the control of cell cycle progression from the G2 phase to the M phase and is also assumed to be involved in the formation and development of tumors. The amount of cyclin B1 in malignant cells is much higher than in normal tissues. Cyclin D1, on the other hand, plays a key role in the regulation of proliferation, connecting extracellular signaling to cell cycle progression. Cyclin D1 expression levels are highly sensitive to proliferative signals. The main role of cyclin D1 in promoting entry into the cell cycle suggests that it should also be important in regulating cell cycle progression once it has begun, although this aspect of its function has been poorly studied due to the difficulty in examining cells that are actively circulating [24]. Despite reports of the deleterious effects of increasing cyclin B1 levels in cancer cells, in our work, we have shown that cell cycle arrest is accompanied by an increase in cyclin B levels. As can be seen in Figure 4, after treatment with a fisetin derivative at a concentration equal to the IC_50_ value, its concentration increases from 14.23% to 21.31%, 12.75% to 25.63% and 11.39% to 20.01% for cells of the A-253, Cal-27 and UCSF-OT-1109 lines, respectively.

We have shown that propargyloxy derivative of fisetin increases the expression of cyclin B1 protein in cells of lines A-253, Cal-27 and UCSF-OT-1109. Therefore, we decided to verify whether there are any changes in the level of cyclin D1. After cytometric analysis, no changes were shown in cyclin D1 levels in the cell lines tested.

### 3.5. Fisetin Derivative Induces Apoptosis in HNC Cells

Since the fisetin derivative exhibited cytotoxic effects on the cancer cells tested, we assessed whether this was associated with induction of apoptosis. Our study showed that exposure to the fisetin derivative induced apoptosis in a dose-dependent manner (Figure 5).

The treatment of cells with the fisetin derivative increased the population of annexin V (+) cells in all cells tested. However, the strongest effect was observed when the fisetin derivative was applied at an IC_50_ concentration against A-253 and FaDu cells. A strong effect was also observed against the Cal-27 line.

As shown in Figure 6, the percentage of annexin (+) cells increased on average from 4.26 ± 0.87% to 48.91 ± 4.98% for the A-253 line when the IC_50_ concentration was applied. The smallest changes were observed for the Detroit 562 line and were not statistically significant. These results coincide with those obtained in the cell cycle analysis. A statistically significant change was also observed for this cell line using a concentration of ¼ IC_50_ where the percentage of cells increased to 13.73 ± 2.69%. In the case of the Cal-27 line, an increase in the percentage of apoptotic cells was observed after applying a concentration equal to the IC_50_ value and ½ IC_50_. In the control sample, the percentage of positive cells was 3.90 ± 0.97%, and increased to 12.78 ± 2.97% and 32.67 ± 5.97% after treatment with the fisetin derivative for ½ IC_50_ and IC_50_ concentrations, respectively. No statistically significant changes in the percentage of apoptotic cells were observed for the Detroit 562 line. Nevertheless, an upward trend was noticeable with increasing concentrations of the fisetin derivative. Smaller changes were observed in relation to cells of the FaDu line. In this case, only after applying the highest concentration tested (IC_50_), statistically significant changes were observed in the percentage of apoptotic cells, which increased more than two times from a value of 14.44 ± 2.67% to a value of 37.98 ± 6.64%. In the case of tongue squamous cell carcinoma cells of the UCSF-OT-1109 line, statistically significant changes in the percentage of apoptotic cells were observed after treatment with the fisetin derivative. Thus, in the case of ½ IC_50_ concentration, the percentage increased from 4.00 ± 0.26% to 27.84 ± 2.06% and to 20.36 ± 4.23% in the case of a higher concentration, i.e., corresponding to the IC_50_ value.

## 4. Discussion

Knowledge of the effects of flavonols on HNC cells is insufficient, despite growing scientific evidence. The modification of natural products and their derivatives is predicted to be crucial for the development of new anticancer drugs. Flavonols induce the apoptosis of cancer cells, including oral cancers. Fisetin exhibits promising chemopreventive activities that include antiproliferative activity, the inhibition of tumor growth by affecting cell cycle regulators and associated apoptosis. The properties of naturally derived compounds that induce apoptosis in cancer cells are key to the formulation of a novel chemopreventive agent. Numerous reports have shown that fisetin induces cytotoxic effects on human cell lines: pancreatic [25], skin [26], breast [27], colorectal [28], oral [29], lung [30], ovarian [31], gastric [32], cervical [33], melanoma [34], osteosarcoma [35], renal [36], leukemia [37] and glioma [38].

As we have shown in previous studies, fisetin inhibits proliferation, reduces cell migration capacity and induces apoptosis in SCC-9, SCC-25 and A-253 HNC cells in a dose-dependent manner, with IC_50_ values of 38.85 µM, 62.34 µM and 49.21 µM, respectively. The results showed that fisetin exposure reduces the expression of Bcl-2 protein while leading to the arrest of the G2/M and S phases of the cell cycle. In addition, we showed that fisetin inhibits cell proliferation by disrupting the cell cycle, which is strongly associated with cycle arrest in the G2/M phase and induces apoptosis by activating caspase-3 and releasing cytochrome c in human HNC cells. In addition, fisetin inhibits anti-apoptotic proteins of the Bcl-2 family, damages the transmembrane potential of mitochondria and increases the level of cytochrome c [8].

The next step in our research was to obtain a fisetin derivative and study its anticancer activity. Several studies confirm that ring substitution plays a modulating role. Studies have suggested that the backbone structure of flavonoids is an important factor in their activity, and the position and degree of hydroxylation are critical. Studies indicate that the 3-hydroxyl, 3′-hydroxyl and 4′-hydroxyl groups of fisetin were crucial for the biological activity of fisetin, while 7-hydroxyl showed negligible activity [39]. The derivative we obtained contains propargyloxy groups at the 3′ and 4′ positions of the B ring, at position 3 of the C ring and position 7 of the A ring. Therefore, we expected a greater increase in the biological activity of the obtained derivative than the starting compound, which was fisetin. The first step was to determine the IC_50_ values of the fisetin derivative against normal cells of the Beas-2B line and cancer cells of the Cal-27, FaDu, A-253, Detroit 562 and UCSF-OT-1109 lines. The selectivity coefficient values were above three in four of the five cell lines. Only in the case of the cells of the A-253 line was the value 1.08, suggesting that the safety of using such a drug is very low, but we proceeded with further studies using all cancer cell lines.

One of the characteristic features of cancer cells is the disruption and abnormal regulation of the cell cycle. A number of chemotherapeutic substances influence the inhibition of neoplastic cell proliferation by inducing cell cycle arrest. Strict regulation through cell cycle control mechanisms is essential to produce two genetically identical cells. Cell cycle checkpoints act on the DNA surveillance principle, preventing the replication of genetic errors and their accumulation in descendant cells. Action is based on delaying the progression of the cell cycle or inducing its termination or cell death in response to irreversible damage to genetic material [40].

Numerous studies have confirmed that fisetin induces G2/M phase cell cycle arrest and apoptosis in cells of cervical cancer [33] and oral cancer, among others [41]. Accordingly, we investigated the effect of a fisetin derivative on cell cycle arrest. In our study, we showed that the fisetin derivative inhibits the cell cycle in the G2/M phase, which was particularly evident in the case of cells of the A-253, Cal-27 and UCSF-OT-1109 lines. Interestingly, we also observed cell cycle inhibition in the S phase in the case of cells of the Cal-27 line. Moreover, in our study, we showed the appearance of cells in the sub-G1 phase of the cell cycle in the case of the A-253 line. In the case of cells of the Detroit 562 line, we did not show the effect of a fisetin derivative for cell cycle inhibition.

Cyclin B1 overexpression has been reported in breast cancer, cervical cancer, gastric cancer, colorectal cancer, SCC of the HNC and non-small cell lung cancer, and its upregulation is closely associated with poor prognosis in various cancer types. The mechanisms responsible for cyclin B1 overexpression are not yet completely understood. Highly expressed cyclin B1, even in the G1 phase, binds to its partner Cdk1, which phosphorylates a number of substrates regardless of the phase of the cell cycle and contributes to aggressive proliferation in tumor tissues. Deregulation of cyclin B1 is involved in tumor transformation and promotes cancer cell proliferation. Downregulation of cyclin B1 can block the aggressive proliferation of cancer cells [42]. In our work, we showed that cell cycle arrest in the G2/M phase induced by a fisetin derivative was accompanied by an increase in cyclin B levels in A-253, Cal-27 and UCSF-OT-1109 lines. However, some studies indicate that after treatment of cells with nocodazole and paclitaxel, a marked increase in cyclin B1 and Cdc2 protein levels was observed [43]. The functional role of this puzzling increase in cyclin B1 protein levels is currently not well understood and requires further study.

The prolonged and inappropriate activation of Cdk1/cyclin B1 mediates proapoptotic signaling in response to mitosis arrest. The reason for this is indirect phosphorylation along with the inactivation of Bcl-2 family proteins, in which the cdk1/cyclin B1 complex is involved. Due to the complexity of this, it also phosphorylates other substrates including caspase-9, which has protective properties against apoptosis of cells during arrested mitosis [44]. Cyclin B1 and cdc2 are also regulators of apoptosis. Apoptosis occurs in response to DNA damage. In addition, apoptosis regulates the cytotoxicity of anticancer drugs, γ-radiation, adriamycin, 5-fluorouracil, etoposide and cisplatin. Apoptosis induced by granzyme B, taxol, Fas and camptothecin is associated with cdc2 activation and/or cyclin B1 protein accumulation. Thus, cdc2 and cyclin B1 are likely involved in multiple apoptotic pathways. In addition, the overexpression of cyclin B1 protein regulates γ-ray-induced apoptosis. Cyclin B1 protein levels increase sharply during γ-radiation-induced apoptosis. Results from other researchers suggest that a cell’s decision to enter apoptosis is regulated, at least in part, by the abundance of cyclin B1 [45]. Studies have also confirmed that the treatment of human prostate cancer cells with 2-methoxyestradiol or docetaxel causes cyclin B1 protein accumulation and an increase in cyclin B1 kinase activity, followed by induction of apoptotic cell death. A positive correlation between cyclin B1 protein and chemotherapy-induced apoptosis in prostate cancer cells has been demonstrated [46].

The next step of the work was to evaluate the effect of a fisetin derivative on the activation of apoptosis in head and neck cancer cells. The study showed that treatment of cells with a fisetin derivative increased the population of annexin V (+) cells in all tested cancer cell lines. However, the strongest effect was observed in A-253, Cal-27 and FaDu cell lines.

A study by Shih et al. [47] demonstrated fisetin-induced apoptotic cell death by increasing reactive oxygen species and Ca^2+^, but decreasing mitochondrial membrane potential and increasing caspase-8, -9 and -3 activity in HSC3 cells. In addition, it was shown that fisetin increased the expression of proapoptotic proteins such as BAK and BAX, but decreased anti-apoptotic proteins such as BCL2 and BCL-x, and increased cleaved forms of caspase-3, -8 and -9 and cytochrome c. Studies by other researchers also confirm that fisetin induces cell death through morphological changes, causes cycle arrest in the G2/M phase, induces apoptosis, promotes ROS and Ca^2+^ production, and decreases the level of ΔΨm and increases the activity of caspase-3, -8 and -9 in SCC-4 cells [41]. In view of the above, it is necessary to know the exact mechanism of action of the obtained fisetin derivative in order to test the safety of its use as an anticancer drug.

## 5. Conclusions

Our study demonstrated that a fisetin derivative inhibited HNC cellular growth and proliferation by affecting important signaling pathways involved in tumor cell growth and differentiation.

The main findings in this study are as follows: (a) propargyloxy fisetin derivative shows high cytotoxicity against Cal-27, FaDu, Detroit 562 and UCST-OT-1109 cell lines; (b) fisetin derivative decreases the cell population in G0/G1 phase and correspondingly increases the cell population in G2/M phase for A-253 cell lines, Cal-27 and UCSF-OT-1109; (c) the appearance of cells in the sub-G1 phase of the cell cycle was observed for line A-253; (d) propargyloxy fisetin derivative increases cyclin B1 protein expression in cells of line A-253, Cal-27 and UCSF-OT-1109; (e) the fisetin derivative does not affect cyclin D1 expression; and (f) the treatment of cells with a fisetin derivative increases the population of annexin-V (+) cells in all cell lines tested. The data indicate that the obtained fisetin derivative exhibits anticancer activity by affecting the cell cycle and increasing apoptosis in selected HNC lines, suggesting its potential use as a new medicinal agent in the future.

## Figures and Tables

**Figure 1 cells-12-02288-f001:**
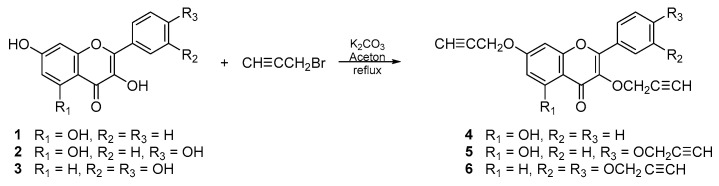
The synthesis route of propargyloxy derivatives of flavone.

**Figure 2 cells-12-02288-f002:**
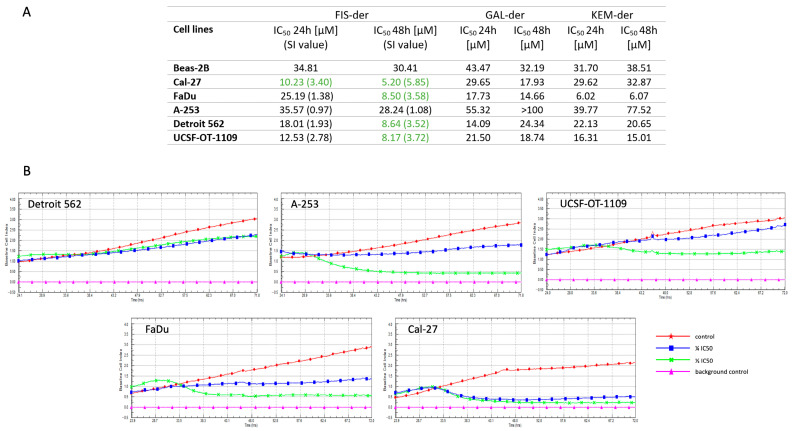
Cytotoxic and antiproliferative effect of fisetin derivative on Detroit 562, A-253, UCSF-OT-1109, FaDu and Cal-27 cell lines. The calculated IC_50_ values of propargyloxy derivatives (fisetin, galangin and kaempferol) for 24 and 48 h of cell incubation are presented in Table. Compounds with *SI* index ≥ 3 are marked in green (**A**). RTCA measurement of cell index values during cancer cells incubation with fisetin derivative (**B**).

**Figure 3 cells-12-02288-f003:**
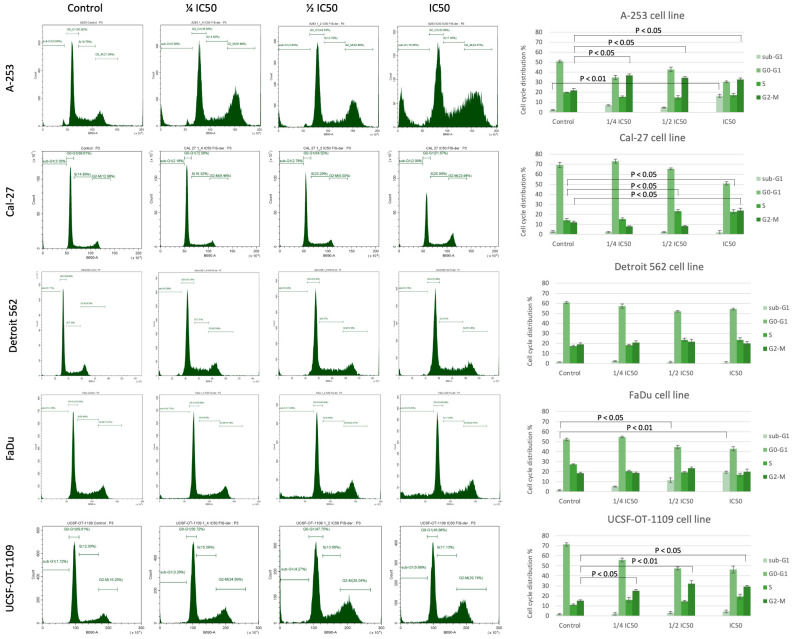
Cell cycle analysis via flow cytometry: Analysis of DNA content in cells treated with fisetin derivative for 24 h was compared with control cells. The analysis showed that the fisetin derivative induces cell cycle arrest in the G0/G1 phase in cells of the A-253, Cal-27 and UCSF-OT-1109 lines. The right side of the figure shows a bar graph depicting quantitative values of flow cytometry data. The data shown are representative of a single experiment. The graphs show the mean ± SD from three independent experiments. Statistical analysis was performed using Student’s *t*-test.

**Figure 4 cells-12-02288-f004:**
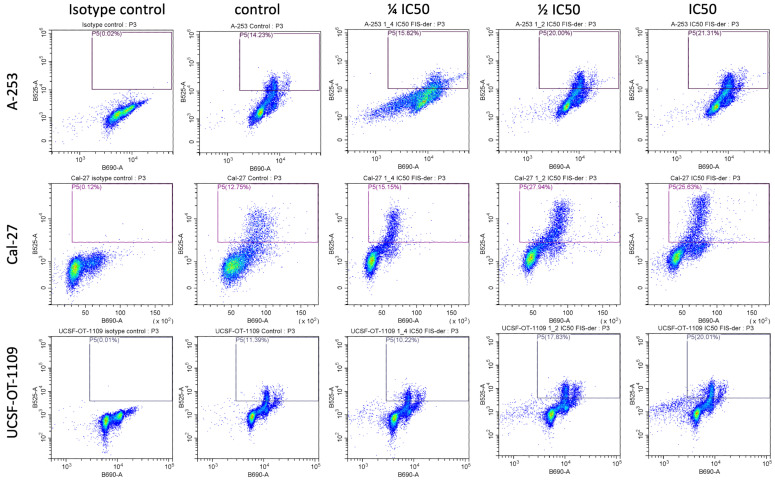
Fisetin derivative enhances cyclin B1 protein expression in cells of A-253, Cal-27 and UCSF-OT-1109 lines. Data shown are representative of a single experiment.

**Figure 5 cells-12-02288-f005:**
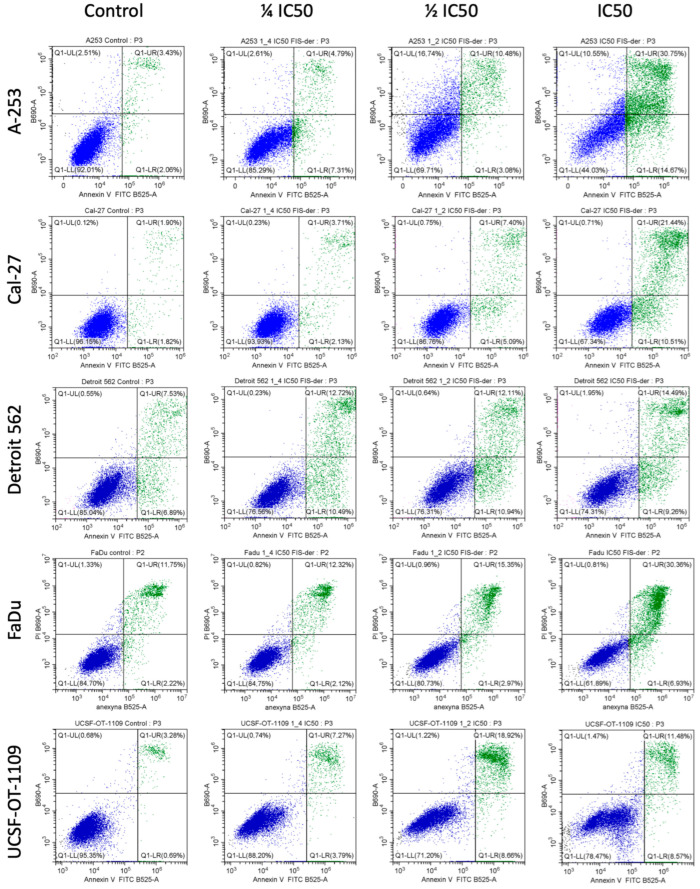
Representative scoring plots of PI (*y*-axis) against annexin V (*x*-axis). Representative scoring plots of PI (*y*-axis) against annexin V (*x*-axis). Variable effects of fisetin derivative on apoptosis induction are noted. Lower left quadrant (FITC−/PI−) = viable cells, lower right quadrant (FITC+/PI−) = early apoptotic cells, upper right quadrant (FITC+/PI+) = late apoptotic cells and upper left quadrant (FITC−/PI+) = necrotic cells.

**Figure 6 cells-12-02288-f006:**
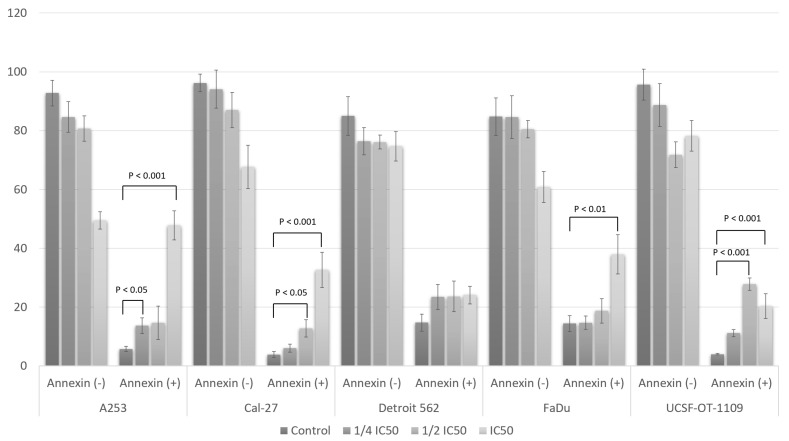
Apoptosis assay using flow cytometry after staining with annexin V-FITC/propidium iodide (PI). Percentage of annexin (−) and annexin (+) cells. Data are presented as the means ± SD of triplicate experiments. Statistical analysis was performed using Student’s *t*-test.

**Table 1 cells-12-02288-t001:** The proton–carbon correlations (HSQC and HMBC experiments) for compound **4** (δ [ppm]-chemical shift of the corresponding signals in the ^1^H NMR and ^13^C NMR spectra).

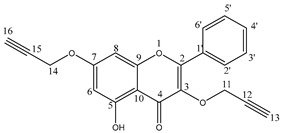					
Proton	^1^H NMR δ [ppm]	HSQC	Carbon	^13^CNMR δ [ppm]	HMBC
-	-	-	C2	136.8	C2(136.8)-H11(4.89)
			C3	131.1	C3(131.1)-H2′(7.46) C3(131.1)-H6′(7.46)
-	-	-	C4	178.8	C4(178.8)-H8(6.66) C4(178.8)-H6(6.32)
OH	12.41				OH(12.41)-C6(98.4) OH(12.41)-C7(106.0) OH(12.41)-C10(161.9)
			C5	156.9	C5(156.9)-H8(6.66)
H6	6.32	H6(6.32)-C6(98.4)	C6	98.4	H6(6.32)-C8(93.2) H6(6.32)-C7(106.0) H6(6.32)-C10(161.9) H6(6.32)-C9(163.8) H6(6.32)-C4(178.8) C6(98.4)-H8(6.66) C6(98.4)-H5(12.41)
			C7	106.0	C7(106.0)-H8(6.66) C7(106.0)-H5(12.41) C7(106.0)-H6(6.32)
H8	6.66	H8(6.66)-C8(93.2)	C8	93.2	H8(6.66)-C6(98.4) H8(6.66)-C7(106.0) H8(6.66)-C5(156.9) H8(6.66)-C9(163.8) H8(6.66)-C4(178.8) C8(93.2)-H6(6.32)
-	-	-	C9	163.8	C9(163.8)-H8(6.66) C9(163.8)-H6(6.32) C9(163.8)-H14(4.84)
-	-	-	C10	161.9	C10(161.9)-H5(12.41) C10(161.9)-H6(6.32)
H11	4.89	H11(4.89)-C11(59.1)	C11	59.1	C11(59.1)-H13(2.88) H11(4.89)-C12(78.3) H11(4.89)-C2(136.8)
-	-	-	C12	78.3	C12(78.3)-H11(4.89)
H13	2.88	H13(2.88)-C13(77.4)	C13	77.4	H13(2.88)-C11(59.1)
H14	4.84	H14(4.84)-C14(56.2)	C14	56.2	C14(56.2)-H16(3.09) H14(4.84)-C15(77.8) H14(4.84)-C9(163.8)
-	-	-	C15	77.8	C15(77.8)-H14(4.84)
H16	3.09	H16(3.09)-C16(77.1)	C16	77.1	H16(3.09)-C14(56.2)
-	-	-	C1′	157.0	C1′(157.0)-H2′(7.46) C1′(157.0)-H6′(7.46) C1′(157.0)-H3′(8.03) C1′(157.0)-H5′(8.03)
H3′	8.03	H2′(8.28)-C3′(128.8)	C3′	128.8	H3′(8.03)-C5′(128.8) H3′(8.03)-C4′(128.5) H3′(8.03)-C1′(157.0) C3′(128.8)-H5′(8.03)
H2′	7.46	H2′(7.46)-C2′(130.5)	C2′	130.5	H2′(7.46)-C6′(130.5) H2′(7.46)-C3(131.1) H2′(7.46)-C1′(157.0) C2′(130.5)-H4′(7.45) C2′(130.5)-H6′(7.46)
H4′	7.45	H4′(7.45)-C4′(128.5)	C4′	128.5	H4′(7.45)-C2′(130.5) H4′(7.45)-C6′(130.5) C4′(128.5)-H3′(8.03) C4′(128.5)-H5′(8.03)
H6′	7.46	H6′(7.46)-C6′(130.5)	C6′	130.5	H6′(7.46)-C2′(130.5) H6′(7.46)-C3(131.1) H6′(7.46)-C1′(157.0) C6′(130.5)-H2′(7.46) C6′(130.5)-H4′(7.45)
H5′	8.03	H5′(8.03)-C5′(128.8)	C5′	128.8	H5′(8.03)-C3′(128.8) H5′(8.03)-C4′(128.5) H5′(8.03)-C1′(157.0) C5′(128.8)-H3′(8.03)

**Table 2 cells-12-02288-t002:** The proton–carbon correlations (HSQC and HMBC experiments) for compound **5** (δ [ppm]-chemical shift of the corresponding signals in the ^1^H NMR and ^13^C NMR spectra).

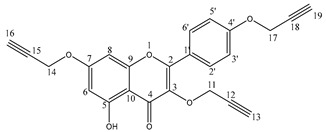					
Proton	^1^H NMR δ [ppm]	HSQC	Carbon	^13^CNMR δ [ppm]	HMBC
-	-	-	C2	136.1	C2(136.1)-H11(4.87)
		-	C3	123.4	C3(123.4)-H2′(7.04) C3(123.4)-H6′(7.04)
-	-	-	C4	178.6	C4(178.6)-H8(6.64) C4(178.6)-H6(6.28) C4(178.6)-H5(12.45)
OH	12.45	-	-	-	OH(12.45)-C6(98.4) OH(12.45)-C7(105.9) OH(12.45)-C10(161.9) OH(12.45)-C4(178.6)
-	-	-	C5	156.8	C5(156.8)-H8(6.64)
H6	6.28	H8(6.28)-C6(98.4)	C6	98.4	H6(6.28)-C8(93.2) H6(6.28)-C7(105.9) H6(6.28)-C10(161.9) H6(6.28)-C9(163.6) H6(6.28)-C4(178.6) C6(98.4)-H8(6.64) C6(98.4)-H5(12.45)
		-	C7	105.9	C7(105.9)-H8(6.64) C7(105.9)-H5(12.45) C7(105.9)-H6(6.28)
H8	6.64	H8(6.64)-C8(93.2)	C8	93.2	H8(6.64)-C6(98.4) H8(6.64)-C7(105.9) H8(6.64)-C5(156.8) H8(6.64)-C9(163.6) H8(6.64)-C4(178.6) C8(93.2)-H6(6.28)
-	-	-	C9	163.6	C9(163.6)-H14(4.81) C9(163.6)-H6(6.28) C9(163.6)-H8(6.64)
-	-	-	C10	161.9	C10(161.9)-H8(6.28) C10(161.9)-H5(12.45)
H11	4.87	H11(4.87)-C11(59.0)	C11	59.0	H11(4.87)-C12(78.3) H11(4.87)-C2(136.1) C11(59.95)-H13(2.87)
-	-	-	C12	78.3	C12(78.3)-H11(4.87)
H13	2.87	H13(2.87)-C13(77.3)	C13	77.3	H13(2.87)-C11(59.0)
H14	4.81	H14(4.81)-C14(56.2)	C14	56.2	H14(4.81)-C15(77.9) H14(4.81)-C9(163.6) C14(56.2)-H16(3.06)
-	-	-	C15	77.9	C15(77.9)-H14(4.81)
H16	3.06	H16(3.06)-C16(77.1)	C16	77.1	H16(3.06)-C14(56.2)
H17	4.79	H17(4.79)-C17(55.6)	C17	55.6	H17(4.79)-C18(78.4) H17(4.79)-C4′(159.9) C17(55.6)-H19(3.02)
-	-	-	C18	78.4	C18(78.4)-H17(4.79)
H19	3.02	H19(3.02)-C19(76.7)	C19	76.7	H19(3.02)-C17(55.6)
-	-	-	C1′	156.7	C1′(156.7)-H3′(8.05) C1′(156.7)-H5′(8.05) C1′(156.7)-H2′(7.04) C1′(156.7)-H6′(7.04)
H2′	7.04	H3′(7.04)-C2′(114.8)	C2′	114.8	H2′(7.04)-C6′(114.8) H2′(7.04)-C3′(130.5) H2′(7.04)-C4′(159.9) H2′(7.04)-C1′(156.7) H2′(7.04)-C3(123.4) C2′(114.8)-H3′(8.05) C2′(114.8)-H6′(7.04)
H3′	8.05	H3′(8.05)-C3′(130.5)	C3′	130.5	H3′(8.05)-C2′(114.8) H3′(8.05)-C5′(130.5) H3′(8.05)-C1′(156.7) H3′(8.05)-C4′(159.9) C3′(130.5)-H2′(7.04) C3′(130.5)-H5′(8.05)
-	-	-	C4′	159.9	C4′(159.9)-H17(4.79) C4′(159.9)-H2′(7.04) C4′(159.9)-H6′(7.04) C4′(159.9)-H3′(8.05) C4′(159.9)-H5′(8.05)
H5′	8.05	H5′(8.05)-C5′(130.5)	C5′	130.5	H5′(8.05)-C6′(114.8) H5′(8.05)-C3′(130.5) H5′(8.05)-C1′(156.7) H5′(8.05)-C4′(159.9) C5′(130.5)-H3′(8.05) C5′(130.5)-H6′(7.04)
H6′	7.04	H6′(7.04)-C6′(114.8)	C6′	114.8	H6′(7.04)-C2′(114.8) H6′(7.04)-C5′(130.5) H6′(7.04)-C4′(159.9) H6′(7.04)-C1′(156.7) H6′(7.04)-C3(123.4) C6′(114.8)-H2′(7.04) C6′(114.8)-H5′(8.05)

## Data Availability

The authors declare that the data supporting the findings of this study are available within the article and its Appendix A, or are available upon request.

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
