# Peer review of "New Propargyloxy Derivatives of Galangin, Kaempferol and Fisetin—Synthesis, Spectroscopic Analysis and In Vitro Anticancer Activity on Head and Neck Cancer Cells"

_cells, 2023, doi:10.3390/cells12182288_

Round 1
Reviewer 1 Report
1. When describing NMR spectra for multiplet signals, intervals of values should be given (lines 125, 126, 134, 142).
2. The J value (line 143) must be given to the nearest tenth (for example 9.0 Hz).
3. Subscripts (lines 121, 129, 138, 146, 234) and superscripts (lines 504, 510) should be observed in formulas.
4. Acetone-d6 is better to replace with acetone-d6 (lines 107, 124, 132, 135, 140, 525, 527, 528, 531).
5. Figure 1: in structures 4-6, R2 and R3 substituents are reversed compared to structures 1-3.
6. Potassium carbonate is insoluble in acetone (lines 117-118). Therefore, it is better to indicate that it has been suspended.

Minor editing of English language required.
Author Response
Re: Research Article cells-2592073
"New Propargyloxy Derivatives of Galangin, Kaempferol and Fisetin – Synthesis, Spectroscopic Analysis and In Vitro Anticancer Activity on Head and Neck Cancer Cells"
Authors: Robert Kubina, Kamil Krzykawski, Arkadiusz Sokal, Marcel Madej, Arkadiusz Dziedzic and Monika Kadela-Tomanek
Status: ' Major Revision Required '
Dear
Lavender Yang
Editor
The authors greatly appreciate valuable and positive comments provided by Reviewers which enhanced our manuscript. We have submitted a revised version of our manuscript "New Propargyloxy Derivatives of Galangin, Kaempferol and Fisetin – Synthesis, Spectroscopic Analysis and In Vitro Anticancer Activity on Head and Neck Cancer Cells". In order to prepare a corrected version, we have followed the comments made by the reviewers. We hope that our revisions will be found satisfactory and the manuscript will be suitable for publication.
Please find our answers and acknowledgements to the Reviewers attached below.
Point-by-point answers to the Reviewer 1:
We would like to express our gratitude for the detailed and critical reading of the manuscript. The authors are convinced that the suggested changes will enhance the quality of the presented work. We truly believe that the Reviewer will find our revisions satisfactory with a manuscript suitable for publication. We appreciate your constructive comments and helpful suggestions.
- When describing NMR spectra for multiplet signals, intervals of values should be given (lines 125, 126, 134, 142).
Authors’ answer: We have corrected all errors indicated by the Reviewer. Intervals of value were added. Thank you.
- The J value (line 143) must be given to the nearest tenth (for example 9.0 Hz).
Authors’ answer: We have corrected the J value. Thank you.
- Subscripts (lines 121, 129, 138, 146, 234) and superscripts (lines 504, 510) should be observed in formulas.
- Acetone-d6 is better to replace with acetone-d6 (lines 107, 124, 132, 135, 140, 525, 527, 528, 531).
- Figure 1: in structures 4-6, R2 and R3 substituents are reversed compared to structures 1-3.
- Potassium carbonate is insoluble in acetone (lines 117-118). Therefore, it is better to indicate that it has been suspended.
Authors’ answer: We have corrected all errors indicated by the Reviewer. Thank you.
All changes suggested by the Reviewer were introduced to the corrected version of the text.
Authors made the corrections based on best understanding of Reviewers recommendations and we do express a sincere hope that our effort fullfilled entirely Reviewers suggestions.
Kind regards
Robert Kubina and co-authors
Reviewer 2 Report
The manuscript has scientific merit however needs improvements mainly in writing. For example, how were the base flavonoids: fisetin, galangin and kaempferol obtained? Were they isolated? Were purchased? In item 3.2 (materials and methods), which controls are used in the experiments? If kaempferol and fisetin derivatives precipitated why another solubilization method was not tested? In addition, the conclusions were too superficial, just a summary of the experiments carried out. Finally, English needs to be improved so that the reader can read the work more fluidly.

The English Language must be improved. I strongly recommend an English review by a native speaker or by services of English edition.
Author Response
Re: Research Article cells-2592073
"New Propargyloxy Derivatives of Galangin, Kaempferol and Fisetin – Synthesis, Spectroscopic Analysis and In Vitro Anticancer Activity on Head and Neck Cancer Cells"
Authors: Robert Kubina, Kamil Krzykawski, Arkadiusz Sokal, Marcel Madej, Arkadiusz Dziedzic and Monika Kadela-Tomanek
Status: ' Major Revision Required '
Dear
Lavender Yang
Editor
The authors greatly appreciate valuable and positive comments provided by Reviewers which enhanced our manuscript. We have submitted a revised version of our manuscript "New Propargyloxy Derivatives of Galangin, Kaempferol and Fisetin – Synthesis, Spectroscopic Analysis and In Vitro Anticancer Activity on Head and Neck Cancer Cells". In order to prepare a corrected version, we have followed the comments made by the reviewers. We hope that our revisions will be found satisfactory and the manuscript will be suitable for publication.
Please find our answers and acknowledgements to the Reviewers attached below.
Point-by-point answers to the Reviewer 2:
We are grateful for your positive opinion regarding our study and presented manuscript content which is greatly appreciated and motivates our team to continue our research activity. We sincerely believe that following some amendments, the Reviewer would find our manuscript satisfactory, and the manuscript can be suitable for publication. Thank you once again.
- For example, how were the base flavonoids: fisetin, galangin and kaempferol obtained? Were they isolated? Were purchased?
Authors’ answer: According to reviewer's suggestion, we added more information about fisetin, galangin and kaempferol in Section 3.2:
“Fisetin (ref. No. PHL82542), galangin (ref. No. 92342) and kaempferol (ref. No. 60010) and other chemicals were purchased from Merck (Darmstadt, German)”.
- In item 3.2 (materials and methods), which controls are used in the experiments?
Authors’ answer: According to reviewer's suggestion, we added more information about controls in Section 3.2:
“All propargyloxy derivatives obtained were evaluated for their in vitro cytotoxicity against five human head and neck region cancer cell lines and one normal bronchial epithelial cell (Beas-2B) using the MTT assay. An additional control consisted of cells cultured in a medium containing an equivalent amount of DMSO without the addition of flavonol-derivative”.
- If kaempferol and fisetin derivatives precipitated why another solubilization method was not tested?
Authors’ answer: The authors did not observe precipitation of galangin and kaempferol derivatives in DMSO solution. Nevertheless, the appearance of crystals was observed when the derivatives solution was added to the culture medium in the MTT assay. Currently, we have no way of determining the cause of the crystal formation. DMSO is not a good solvent for crystallization of any compound though it dissolves many compounds. Crystal structures were grown from a mixture of DMSO and medium (water). Our goal is to further study the obtained derivatives. However, at the beginning we intend to carry out chemical analyzes of the obtained crystalline forms in order to select the appropriate conditions for cell culture with the tested compounds.
- In addition, the conclusions were too superficial, just a summary of the experiments carried out.
Authors’ answer: According to Reviewer's suggestion, new conclusions were included to the manuscript and updated.
- Finally, English needs to be improved so that the reader can read the work more fluidly.
Authors’ answer: English proofreading of our article was performed by English native speaker (MDPI service).
All changes suggested by the Reviewer were introduced to the corrected version of the text.
Authors made the corrections based on best understanding of Reviewers recommendations and we do express a sincere hope that our effort fullfilled entirely Reviewers suggestions.
Kind regards
Robert Kubina and co-authors